# Learning compositional functions via multiplicative weight updates

**Jeremy Bernstein**
Caltech

**Jiawei Zhao**
Caltech

**Markus Meister**
Caltech

**Ming-Yu Liu**
NVIDIA

**Anima Anandkumar**
Caltech / NVIDIA

**Yisong Yue**
Caltech

## Abstract

Compositionality is a basic structural feature of both biological and artificial neural networks. Learning compositional functions via gradient descent incurs well known problems like vanishing and exploding gradients, making careful learning rate tuning essential for real-world applications. This paper proves that multiplicative weight updates satisfy a descent lemma tailored to compositional functions. Based on this lemma, we derive *Madam*—a multiplicative version of the Adam optimiser—and show that it can train state of the art neural network architectures without learning rate tuning. We further show that Madam is easily adapted to train natively compressed neural networks by representing their weights in a logarithmic number system. We conclude by drawing connections between multiplicative weight updates and recent findings about synapses in biology.

## 1 Introduction

Neural computation in living systems emerges from the collective behaviour of large numbers of low precision and potentially faulty processing elements. This is a far cry from the precision and reliability of digital electronics. Looking at the numbers, a synapse on a computer is often represented using 32 bits taking more than 4 billion distinct values. In contrast, a biological synapse is estimated to take 26 distinguishable strengths requiring only 5 bits to store [1]. This is a discrepancy between nature and engineering that spans many orders of magnitude. So why does the brain learn stably whereas deep learning is notoriously finicky and sensitive to myriad hyperparameters?

Meanwhile, an industrial effort is underway to scale artificial networks *up* to run on supercomputers and *down* to run on resource-limited edge devices. While learning algorithms designed to run natively on low precision hardware could lead to smaller and more power efficient chips [2, 3], progress is hampered by our poor understanding of how precision impacts learning. As such, existing numerical representations have developed somewhat independently from learning algorithms. The next generation of neural hardware could benefit from more principled algorithmic co-design [4].

**Our contributons:**

1. Building on recent results in the perturbation analysis of compositional functions [5], we show that a multiplicative learning rule satisfies a descent lemma tailored to neural networks.

2. We propose and benchmark *Madam*—a multiplicative version of the Adam optimiser. Empirically, Madam seems to not require learning rate tuning. Further, it may be used to train neural networks with low bit width synapses stored in a logarithmic number system.

3. We point out that multiplicative weight updates respect certain aspects of neuroanatomy. First, synapses are exclusively *excitatory* or *inhibitory* since their sign is preserved under the update. Second, multiplicative weight updates are most naturally implemented in a logarithmic number system, in line with anatomical findings about biological synapses [1].

## 2    Related work

**Multiplicative weight updates**    Multiplicative algorithms have a storied history in computer science. Examples in machine learning include the Winnow algorithm [6] and the exponentiated gradient algorithm [7]—both for learning linear classifiers in the face of irrelevant input features. The Hedge algorithm [8], which underpins the AdaBoost framework for boosting weak learners, is also multiplicative. In algorithmic game theory, multiplicative weight updates may be used to solve two-player zero sum games [9]. Arora et al. [10] survey many more applications.

Multiplicative updates are typically viewed as appropriate for problems where the geometry of the optimisation domain is described by the relative entropy [7], as is often the case when optimising over probability distributions. Since the relative entropy is a Bregman divergence, the algorithm may then be studied under the framework of mirror descent [11]. We suggest that multiplicative updates may arise under a broader principle: when the geometry of the optimisation domain is described by any relative distance measure.

**Deep network optimisation**    Finding the right distance measure to describe deep neural networks is an ongoing research problem. Some theoretical work has supposed that the underlying geometry is Euclidean via the assumption of Lipschitz-continuous gradients [12], but Zhang et al. [13] suggest that this assumption may be too strong and consider relaxing it. Similarly, Azizan et al. [14] consider more general Bregman divergences under the mirror descent framework, although these distance measures are not tailored to compositional functions like neural networks. Neyshabur et al. [15], on the other hand, derive a distance measure based on paths through a neural network in order to better capture the scaling symmetries over layers. Still, it is difficult to place this distance within a formal optimisation theory in order to derive descent lemmas.

Recent research has looked at learning algorithms that make relative updates to network layers, such as LARS [16], LAMB [17], and Fromage [5]. Empirically, these algorithms appear to stabilise large batch network training and require little to no learning rate tuning. Bernstein et al. [5] suggested that these algorithms are accounting for the compositional structure of the neural network function class, and derived a new distance measure called *deep relative trust* to describe this analytically. It is the relative nature of deep relative trust that leads us to propose multiplicative updates in this work.

**Numerics of a synapse**    A basic goal of theoretical neuroscience is to connect the numerical properties of a synapse with network function. For example, following the observation that synapses are exclusively excitatory or inhibitory, van Vreeswijk and Sompolinsky [18] studied how the balance of excitation and inhibition can affect network dynamics and Amit et al. [19] studied perceptron learning with sign-constrained synapses. More recently, based on the observation that synapse size and strength are correlated, Bartol et al. [1] used the number of distinguishable synapse sizes to estimate the information content of a synapse. Their results suggest that biological synapses may occupy just 26 levels in a logarithmic number system, thus storing less than 5 bits of information. This leads to one estimate that a human brain may store no more than:

$$5 \text{ bits per synapse} \times 100 \text{ trillion synapses} \approx 60 \text{ terabytes of data.}$$

**Low-precision hardware**    In their bid to outrun the end of Moore's Law, chip designers have also taken an interest in understanding and improving the efficiency of artificial synapses. This work dates back at least to the 1980s and 1990s—for example, Akira Iwata et al. [20] designed a 24-bit neural network accelerator while Baker and Hammerstrom [2] suggested that learning may break down below 12 bits per synapse. In 1993, Holt and Hwang [21] analysed round-off error for compounding operators and proposed a heuristic connection between numerics and optimisation (their Equation 54).

Last decade there was renewed interest in low precision synaptic weights both for deployment [22–24] and training [25–29] of artificial networks. This research has included the exploration of logarithmic number systems [30, 31]. A general trend has emerged: a trained network may be quantised to just a few bits per synapse, but 8 to 16 bits are typically required for stable learning [25, 27, 28]. Given the lack of theoretical understanding of how precision relates to learning, these works often introduce subtle but significant complexities. For example, existing works using logarithmic number systems combine them with additive optimisation algorithms like Adam and SGD [30], thus requiring tuning of both the learning algorithm *and* the numerical representation. And many works must resort to using high precision weights in the final layer of the network to maintain accuracy [27–29].

# 3 Mathematical model

A basic question in the theory of neural networks is as follows:

> How far can we perturb the synapses before we damage the network as a whole?

In this paper, this question is important on two fronts: our learning rule must not destroy the information contained in the synapses, and our numerical representation must be precise enough to encode non-destructive perturbations. Once it has been established that multiplicative updates are a good learning rule (addressing the first point) it becomes natural to represent them using a logarithmic number system (addressing the second). Therefore, this section shall focus on establishing—first as a sketch, then rigorously—the benefits of multiplicative updates for learning compositional functions.

## 3.1 Sketch of the main idea

The *raison d'être* of a synaptic weight is to support learning in the network as a whole. In machine learning, this is formalised by constructing a loss function $\mathcal{L}(W)$ that measures the error of the network in weight configuration $W$. Learning proceeds by perturbing the synapses $W \to W + \Delta W$ in order to reduce the loss function. A good perturbation direction is the negative gradient of the loss: $\Delta W \propto -\nabla_W \mathcal{L}(W)$. But how far can this direction be trusted?

Intuitively, the negative gradient should only be trusted until its approximation quality breaks down. This breakdown could be measured by the Hessian of the loss function, but this is intractable for large networks since it involves *all pairs* of weights. Instead, Bernstein et al. [5] suggest how to operate without the Hessian. To get a handle on exactly how this is done, consider an $L$ layer multilayer perceptron $f(x)$ and the gradient $g_k(W)$ of its loss with respect to the weights at the $k$th layer $W_k$:

$$g_k(W) := \nabla_{W_k} \mathcal{L}(W) = \frac{\partial \mathcal{L}}{\partial f} \cdot \frac{\partial f}{\partial h_k} \cdot \frac{\partial h_k}{\partial W_k}, \tag{1}$$

where $h_k$ denotes the activations at the $k$th hidden layer of the network. By the backpropagation algorithm [32]—and ignoring the nonlinearity for the sake of this sketch—the second term in Equation 1 depends on the product of weight matrices over layers $k + 1$ to $L$, and the third term depends on the product of weight matrices over layers 1 to $k - 1$. It is therefore natural to model the relative change in the whole expression via the formula for the relative change of a product:

$$\frac{\|g_k(W + \Delta W) - g_k(W)\|_F}{\|g_k(W)\|_F} \sim \frac{\left\|\prod_{l=1}^{L}(W_l + \Delta W_l) - \prod_{l=1}^{L} W_l\right\|_F}{\left\|\prod_{l=1}^{L} W_l\right\|_F} \sim \prod_{l=1}^{L}\left(1 + \frac{\|\Delta W_l\|_F}{\|W_l\|_F}\right) - 1.$$

This neglects the specific choice of loss function which enters via the first term $\partial \mathcal{L}/\partial f$ in Equation 1.

To recap, we have proposed that neural networks ought to be trained by following the gradient of their loss until that gradient breaks down. We then sketched that the relative breakdown in gradient depends on a product over relative perturbations to each network layer. The simplest perturbation that follows the gradient direction *and* keeps the layerwise relative perturbation small was first introduced by You et al. [16]. Letting $\eta$ be a positive scalar known as the *learning rate*, the update is given by:

$$W_k \to W_k - \eta \frac{\|W_k\|_F}{\|g_k\|_F} g_k \quad \text{for each layer } k = 1, ..., L.$$

The downside of this rule is that it requires knowing precisely which weights act together as a layer and normalising those updates jointly. It is difficult to imagine this happening in the brain, and for exotic artificial networks it is sometimes unclear what constitutes a layer [33]. A convenient way to sidestep this issue is to update each individual weight $w$ multiplicatively, via:

$$w \to w(1 \pm \eta) \approx w \mathrm{e}^{\pm \eta} \quad \text{for each weight } w.$$

This update ensures that $\|\Delta W_*\|_F / \|W_*\|_F$ is small *for every subset $W_*$ of the weights $W$* whilst only using information local to a synapse. Therefore, by appropriate choice of the signs of the multiplicative factors, it can be arranged that this perturbation is roughly aligned with the negative gradient whilst keeping the relative breakdown in gradient small.

On the next page, we shall develop this sketch into a rigorous optimisation theory.

## 3.2 First-order optimisation of continuously differentiable functions

To elucidate the connection between gradient breakdown and optimisation, consider the following inequality. Though it applies to all continuously differentiable functions, we will think of $\mathcal{L}$ as a loss function measuring the performance of a neural network of depth $L$ at some task.

**Lemma 1.** *Consider a continuously differentiable function $\mathcal{L} : \mathbb{R}^n \to \mathbb{R}$ that maps $W \mapsto \mathcal{L}(W)$. Suppose that parameter vector $W$ decomposes into $L$ parameter groups: $W = (W_1, W_2, ..., W_L)$, and consider making a perturbation $\Delta W = (\Delta W_1, \Delta W_2, ..., \Delta W_L)$. Let $\theta_k$ measure the angle between $\Delta W_k$ and negative gradient $-g_k(W) := -\nabla_{W_k} \mathcal{L}(W)$. Then:*

$$\mathcal{L}(W + \Delta W) - \mathcal{L}(W) \leq -\sum_{k=1}^{L} \|g_k(W)\|_F \|\Delta W_k\|_F \left[ \cos \theta_k - \max_{t \in [0,1]} \frac{\|g_k(W + t\Delta W) - g_k(W)\|_F}{\|g_k(W)\|_F} \right].$$

The proof is in Appendix A. The result says: *to reduce a function, follow its negative gradient until it breaks down*. Descent is formally guaranteed when the bracketed terms are positive. That is, when:

$$\max_{t \in [0,1]} \frac{\|g_k(W + t\Delta W) - g_k(W)\|_F}{\|g(W)\|_F} < \cos \theta_k \qquad \text{(for } k = 1, ..., L). \tag{2}$$

According to Equation 2, to guarantee descent for neural networks, we must bound their relative breakdown in gradient. To this end, Bernstein et al. [5] propose the notion of *deep relative trust*.

**Modelling assumption 1.** [Deep relative trust] Consider a neural network with $L$ layers and parameters $W = (W_1, W_2, ..., W_L)$. Consider parameter perturbation $\Delta W = (\Delta W_1, \Delta W_2, ..., \Delta W_L)$. Let $g_k(W) := \nabla_{W_k} \mathcal{L}(W)$ denote the gradient of the loss. Then the gradient breakdown is bounded by:

$$\frac{\|g_k(W + \Delta W) - g_k(W)\|_F}{\|g_k(W)\|_F} \leq \prod_{l=1}^{L} \left( 1 + \frac{\|\Delta W_l\|_F}{\|W_l\|_F} \right) - 1 \qquad \text{(for } k = 1, ..., L).$$

While deep relative trust is based on a perturbation analysis of multilayer perceptrons with leaky relu nonlinearity, it may be seen as a model of more general neural networks since the product reflects their compositional structure. Crucially, compared to a Hessian that may contain as many as $10^{18}$ entries for modern networks (since the Hessian's size is quadratic in the number of parameters), deep relative trust gives a tractable analytic model for the breakdown in gradient.

## 3.3 Descent via multiplicative weight updates

Since deep relative trust penalises the relative size of the perturbation to each layer, it is natural that our learning algorithm would bound these perturbations on a relative scale. A simple way to achieve this is via the following *multiplicative* update rule:

$$W \to W + \Delta W = W \odot \left[ 1 - \eta \, \text{sign} \, W \odot \text{sign} \, g(W) \right], \tag{3}$$

where the sign is taken elementwise and $\odot$ denotes elementwise multiplication. Synapses shrink where the signs of $W$ and $g(W)$ agree and grow where the signs differ. In the following theorem, we establish descent under this update for compositional functions described by deep relative trust.

**Theorem 1.** *Let $\mathcal{L}$ be the continuously differentiable loss function of a neural network of depth $L$ that obeys deep relative trust. For $k = 1, ..., L$, let $0 \leq \gamma_k \leq \frac{\pi}{2}$ denote the angle between $|g_k(W)|$ and $|W_k|$ (where $|\cdot|$ denotes the elementwise absolute value). Then the multiplicative update in Equation 3 will decrease the loss function provided that:*

$$\eta < (1 + \cos \gamma_k)^{\frac{1}{L}} - 1 \qquad \text{(for all } k = 1, ..., L).$$

Theorem 1 tells us that for small enough $\eta$, multiplicative updates achieve descent. It also tells us *on what scale $\eta$ must be small*. The proof is given in Appendix A.

We can bring Theorem 1 to life by plugging in numbers. First, we must consider what values the angles $\gamma_k$ are likely to take. Since $|g_k(W)|$ and $|W_k|$ are nonnegative vectors, the angle between them can be no larger than $\pi/2$, which happens when the support of the two vectors is disjoint. A simple model to consider is two high-dimensional Gaussian vectors with iid components, for which the angle between their absolute values satisfies $\cos \gamma_k \approx 0.64$. Taking this value, Theorem 1 implies that for a 40 layer network, setting $\eta = 0.01$ guarantees descent. We find that $\eta = 0.01$ works well in all our experiments with the *Madam* optimiser in later sections.

**Algorithm 1** *Madam*—a multiplicative adaptive moments based optimiser. Good default hyperparameters are: $\eta = 0.01$, $\eta^* = 8\eta$, $\sigma^* = 3\sigma$, $\beta = 0.999$. $\sigma$ can be lifted from a standard initialisation.

**Numerical representation:** initial weight scale $\sigma$; max weight $\sigma^*$.
**Optimisation parameters:** typical perturbation $\eta$; max perturbation $\eta^*$; averaging constant $\beta$.
**Weight initialisation:** initialise weights randomly on scale $\sigma$, for example: $W \sim \text{NORMAL}(0, \sigma)$.

   $\bar{g} \leftarrow 0$           ▷ initialise second moment estimate
   **repeat**
      $g \leftarrow \text{StochasticGradient}()$       ▷ collect gradient
      $\bar{g}^2 \leftarrow (1-\beta)g^2 + \beta\bar{g}^2$       ▷ update second moment estimate
      $W \leftarrow W \odot \exp[-\eta \,\text{sign}\, W \odot \text{clamp}_{\eta^*/\eta}(g/\bar{g})]$    ▷ update weights multiplicatively
      $W \leftarrow \text{clamp}_{\sigma^*}(W)$       ▷ clamp weights between $\pm\sigma^*$
   **until** converged

## 4 Making the algorithm practical—the *Madam* optimiser

In the previous section, we built an optimisation theory for the multiplicative update rule appearing in Equation 3. While that update yields a straightforward mathematical analysis, two modifications render it more practically useful. First, we use the fact that $1 + x \approx \exp x$ for small $x$ to approximate Equation 3 by:

$$W + \Delta W = W \odot \exp\left[-\eta \,\text{sign}\, W \odot \,\text{sign}\, g\right], \tag{4}$$

where $g$ is shorthand for $g(W)$. This change makes it easier to represent weights in a *logarithmic number system*, since the weights are restricted to integer multiples of $\eta$ in log space. Second, in practice it may be overly stringent to restrict to the 1 bit gradient sign. This leads us to propose:

$$W + \Delta W = W \odot \exp\left[-\eta \,\text{sign}\, W \odot \,\text{clamp}_{\eta^*/\eta}\left(\frac{g}{\bar{g}}\right)\right]. \tag{5}$$

In this expression, $\bar{g}$ denotes the root mean square gradient, which is estimated by a running average over iterations. Each iteration, this is updated by:

$$\bar{g}^2 \leftarrow (1-\beta)\,g^2 + \beta\,\bar{g}^2.$$

This idea is borrowed from the Adam and RMSprop optimisers [34, 35]. Since $g/\bar{g}$ should typically be $O(\pm 1)$ (where $O(\cdot)$ means *on the order of*), it may be viewed as a higher precision version of $\text{sign}\, g$. The $\text{clamp}_a(\cdot)$ function projects its argument on to the interval $[-a, a]$. This gradient clamping means that a weight can change by no more than a factor of $\exp(\pm\eta \times \eta^*/\eta) = \exp(\pm\eta^*) \approx 1 \pm \eta^*$ per iteration, ensuring that the algorithm still makes bounded relative perturbations and respects deep relative trust. Finally, we introduce a weight clamping operation—see the full pseudocode in Algorithm 1. As discussed in Section 8, weight clamping *stabilises* and *regularises* the algorithm.

### 4.1 *B*-bit Madam

The multiplicative nature of Madam (Algorithm 1) suggests storing synapse strengths in a logarithmic number system, where numbers are represented just by a sign and exponent. To see why this is natural for Madam consider that, since $g/\bar{g}$ is typically $O(\pm 1)$, Madam's typical relative perturbation to a synapse is $\exp \pm\eta$. Therefore, in log space, the synapse strengths typically change by $\pm\eta$. This suggests efficiently representing a synapse by its sign and an integer multiple of $\eta$.

In practice, a slightly more fine-grained discretisation is beneficial. We define the *base precision* $\eta_0$ which divides both the learning rate $\eta$ and maximum perturbation strength $\eta^*$. We then round $g/\bar{g}$ appearing in Madam to the nearest multiple of $\eta_0/\eta$. This leads to quantised multiplicative updates:

$$e^0, \quad e^{\pm\eta_0}, \quad e^{\pm 2\eta_0}, \quad ..., \quad e^{\pm(\eta^*-2\eta_0)}, \quad e^{\pm(\eta^*-\eta_0)}, \quad e^{\pm\eta^*}.$$

A good setting in practice is $\eta_0 = 0.001$, $\eta = 0.01$ and $\eta^* = 0.08$. Finally, to obtain a $B$-bit weight representation, we must restrict the number of allowed weight levels to $2^B$. The resulting representation is:

$$W = \text{sign}\,(W)\,\sigma^* e^{-k\eta_0} \qquad \text{for } k \in \{0, 1, ..., 2^B - 1\}.$$

The scale parameter $\sigma^*$ is shared by a whole network layer, and may be taken from an existing weight initialisation scheme. A good mental picture is that the weights live on a ladder in log space with rungs spaced apart by $\eta_0$. The Madam update moves the weights up or down the rungs of the ladder.

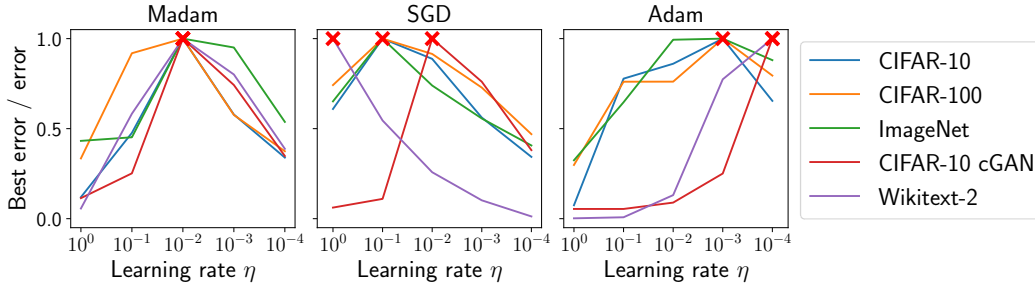

Figure 1: Learning rate tuning. We tuned the learning rate $\eta$ over a logarithmic grid on various deep learning benchmarks. For each run, a fixed $\eta$ is used without decay. For each optimiser and setting of $\eta$, we plot the best test error across all $\eta$ for that optimiser divided by the test error for that specific $\eta$, meaning that a value of $1.0$ (highlighted with a red cross) indicates the best setting of $\eta$ for that optimiser and task. For Madam, the best setting ($\eta = 0.01$) was independent of task.

Table 1: Results after tuning the learning rate $\eta$. For each task, we compare to the better performing optimiser in {SGD, Adam} and list the associated inital $\eta$. We quote top-1 test error, FID [36] and perplexity for the classifiers, GAN and transformer respectively. Lower is better in all cases. The mean and range are based on three repeats. Complete results are given in Table 3 in Appendix B.

*Note: all Madam runs use initial learning rate $\eta = 0.01$.*
*For all algorithms, $\eta$ is decayed by 10 when the loss plateaus.*

| Dataset | Task | Best Baseline | Baseline $\eta$ | Baseline Result | Madam $\eta$ | Madam Result |
|---------|------|---------------|-----------------|-----------------|--------------|--------------|
| CIFAR-10 | Resnet18 | Adam | 0.001 | $6.6 \pm 0.2$ | 0.01 | $7.8 \pm 0.2$ |
| CIFAR-100 | Resnet18 | SGD | 0.1 | $29.1 \pm 0.2$ | 0.01 | $30.2 \pm 0.1$ |
| ImageNet | Resnet50 | SGD | 0.1 | $24.1 \pm 0.1$ | 0.01 | $28.9 \pm 0.1$ |
| CIFAR-10 | cGAN | Adam | 0.0001 | $23.9 \pm 0.9$ | 0.01 | $19 \pm 2$ |
| Wikitext-2 | Transformer | SGD | 1.0 | $169.6 \pm 0.6$ | 0.01 | $173.3 \pm 0.6$ |

## 5   Benchmarking Madam in FP32

In this section, we benchmark Madam (Algorithm 1) with weights represented in 32-bit floating point. In the next section, we shall benchmark $B$-bit Madam. The results in this section show that—across various tasks, including image classification, language modeling and image generation—Madam *without learning rate tuning* is competitive with a *tuned* SGD or Adam.

In Figure 1, we show the results of a learning rate grid search undertaken for Madam, SGD and Adam. The optimal learning rate setting for each benchmark is shown with a red cross. Notice that for Madam the optimal learning rate is in all cases $0.01$, whereas for SGD and Adam it varies.

In Table 1, we compare the final results using tuned learning rates for Adam and SGD and using $\eta = 0.01$ for Madam. Madam's results are competitive, and substantially better in the GAN experiment where SGD obtained FID $> 30$. Madam performed worst on the Imagenet experiment, achieving a test error 4.8% worse than SGD. These results were attained using epoch budgets lifted from the baseline implementations.

**Implementation details**   The code for these experiments is to be found at `https://github.com/jxbz/madam`. Because bias terms are initialised to zero by default in Pytorch [37], a multiplicative update would have no effect on these parameters. Therefore we intialised the biases away from zero in some experiments, which led to slight performance improvements. Madam benefited from light tuning of the $\sigma^*$ hyperparameter (see Algorithm 1) which regularises the network by controlling the maximum size of the weights. In each experiment, $\sigma^*$ was set to be 1 to 5 times larger than the initialisation scale on a layerwise basis. Tuning $\sigma^*$ had a comparable effect on final performance to the effect of tuning weight decay in SGD. More experimental details are given in Appendix B.

Table 2: Benchmarking $B$-bit Madam. We tested 12-bit, 10-bit and 8-bit Madam on various tasks. The results for the FP32 baseline are reproduced from Table 1. For each result we give the mean and range over three repeats. In all experiments, an initial learning rate $\eta$ of $O(0.01)$ was used. In all 12-bit experiments, the base precision was chosen as $\eta_0 = 0.001$. In order to reduce the bit width from 12 to 8 bits, we increased the base precision $\eta_0$ of the numerical representation finding this to work better than the alternative: reducing the dynamic range of the numerical representation.

| Dataset | Task | FP32 Madam | 12-bit | 10-bit | 8-bit |
|---------|------|-----------|--------|--------|-------|
| CIFAR-10 | Resnet18 | $7.8 \pm 0.2$ | $7.0 \pm 0.1$ | $7.8 \pm 0.3$ | $8.6 \pm 1.5$ |
| CIFAR-100 | Resnet18 | $30.2 \pm 0.1$ | $27.6 \pm 0.3$ | $29.5 \pm 0.3$ | $33.9 \pm 1.1$ |
| ImageNet | Resnet50 | $28.9 \pm 0.1$ | $31.1 \pm 0.1$ | $34.8 \pm 0.3$ | $50.5 \pm 0.5$ |
| CIFAR-10 | cGAN | $19.3 \pm 0.7$ | $19.8 \pm 0.8$ | $23.4 \pm 0.4$ | $36 \pm 6$ |
| Wikitext-2 | Transformer | $173.3 \pm 0.6$ | $182.3 \pm 0.6$ | $218.0 \pm 0.6$ | $262 \pm 2$ |

## 6 Benchmarking $B$-bit Madam

The results in this section demonstrate that $B$-bit Madam can be used to train networks that use 8–12 bits per weight, often with little to no loss in accuracy compared to an FP32 baseline. This compression level is in the range of 8–16 bits suggested by prior work [25, 27, 28]. However, we must emphasise the ease with which these results were attained. Just as Madam did not require learning rate tuning (see Figure 1), neither did $B$-bit Madam. In all 12-bit runs, a learning rate of $\eta = 0.01$ combined with a base precision $\eta_0 = 0.001$ could be relied upon to achieve stable learning.

The results are given in Table 2. Though little deterioration is experienced at 12 bits, we believe that the results could be improved by making minor hyperparameter tweaks. For example, in the 12-bit ImageNet experiment we were able to reduce the error from $\sim 29\%$ to $\sim 25\%$ by borrowing layerwise parameter scales $\sigma^*$ (see Section 4.1) from a pre-trained model instead of using the standard Pytorch [37] initialisation scale. Still, it would be against the spirit of this work to present results with over-tuned hyperparameters.

To get more of a feel for the relative simplicity of our approach, we shall briefly comment on some of the subtleties introduced by prior work that $B$-bit Madam avoids. Studies often maintain higher-precision copies of the weights as part of their low-precision training process [28, 29]. For example, in their paper on 8-bit training, Wang et al. [28] actually maintain a 16-bit master copy of the weights. Furthermore, it is common to keep certain network layers such as the output layer at higher precision [27–29]. In contrast, we use the same bit width to represent every layer's weights, and weights are both stored and updated in their $B$-bit representation.

Furthermore, we want to emphasise how natural it is to combine multiplicative updates with a logarithmic number system. Prior research on deep network training using logarithmic number systems has combined them with additive optimisation algorithms like SGD [30]. This necessitates tuning both the number system hyperparameters (dynamic range and base precision) *and* the optimisation hyperparameter (learning rate). As was demonstrated in Figure 1, tuning the SGD learning rate is already a computationally intensive task. Moreover, the cost of hyperparameter grid search grows *exponentially* in the number of hyperparameters.

**Implementation details** The code for these experiments is to be found at `https://github.com/jxbz/madam`. The $B$-bit Madam hyperparameters are defined in Section 4.1. We choose the layerwise scale $\sigma^*$ in $B$-bit Madam following the same strategy as $\sigma^*$ in Madam: 1 to 5 times the default Pytorch [37] initialisation scale, except for biases where the default Pytorch initialisation is zero. We choose the initial learning rate $\eta$ to be $O(0.01)$ across all experiments. In the 12-bit experiments, we choose the base precision $\eta_0 = 0.001$. In the 10-bit and 8-bit experiments, a larger base precision is used—still satisfying $\eta_0 \leq \eta = O(0.01)$. For each layer, we initialise the weights uniformly from:

$$\left\{ \pm \sigma^* e^{-k\eta_0} : k \in \{0, 1, ..., 2^B - 1\} \right\}.$$

Notice that $\eta_0$ sets the precision of the representation, and the dynamic range is given by $\exp\left[(2^B - 1)\eta_0\right]$. For $B = 12$ bits and base precision $\eta_0 = 0.001$ as used in the 12-bit experiments, the dynamic range is 60.0 to 1 decimal place. Finally, during training the learning rate $\eta$ is decayed toward the base precision $\eta_0$ whenever the loss plateaus. See Appendix B for more detail.

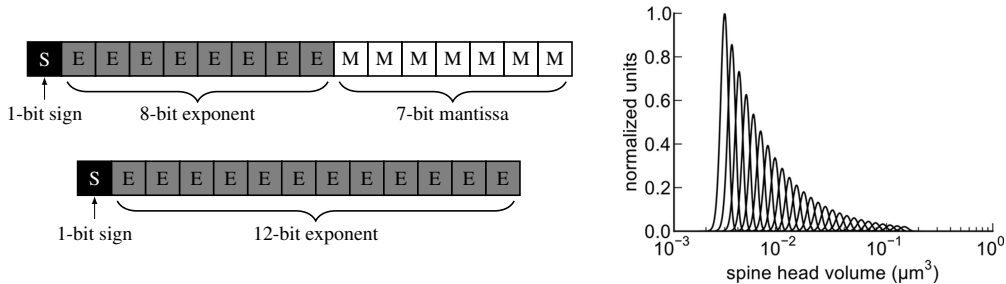

Figure 2: Upper left: the `bfloat16` number system used in Google's TPU chips [38]. Lower left: the logarithmic number system suggested by our theory. Right: the synaptic number system suggested by Bartol et al. [1] based on 3D electron microscope images of hippocampal neuropil in three adult male rats. Synapses are suggested to take 26 distinguishable strengths (believed to be correlated with spine head volume) on a logarithmic number system. This plot is reproduced from [1, Figure 8].

## 7 Discussion

After studying the optimisation properties of compositional functions, we have confirmed that neural networks may be trained via multiplicative updates to weights stored in a logarithmic number system. We shall now discuss possible implications for both chip design and neuroscience.

**Computer number systems** In an effort to accelerate and reduce the cost of machine learning workflows, chip designers are currently exploring low precision arithmetic in both GPUs and TPUs. For example, Google has used `bfloat16`—or *brain floating point*—in their TPUs [38] and NVIDIA has developed a mixed precision GPU training system [39]. A basic question in the design of low-precision number systems is how the bits should be split between the exponent and mantissa. As shown in Figure 2 (left), `bfloat16` opts for an 8-bit exponent and 7-bit mantissa. Our work supports a prior suggestion [30] that to represent network weights, a mantissa may not be needed at all.

Curiously, the same observation is made by Bartol et al. [1] in the context of neuroscience. In Figure 2 (right), we reproduce a plot from their paper that illustrates this. The authors suggest that the brain may use "a form of non-uniform quantization which efficiently encodes the dynamic range of synaptic strengths at constant precision"—or in other words, a logarithmic number system. The authors found that "spine head volumes ranged in size over a factor of 60 from smallest to largest", where spine head volume is a correlate of synapse strength. Coincidentally, $B$-bit Madam with $B = 12$ and base precision $\eta_0 = 0.001$ (as in our experiments) has the same dynamic range of $\exp[(2^B - 1)\eta_0] \approx 60$.

**Frozen signs and Dale's principle** The neuroscientific principle that synapses cannot change sign is sometimes referred to as *Dale's principle* [19]. When compositional functions are learnt via multiplicative updates, the signs of the weights are frozen and can be thought to satisfy Dale's principle. This means that after training with multiplicative updates, there is no need to store the sign bits since they may be regenerated from the random seed. This could also impact hardware design since it would technically be possible to freeze a random sign pattern into the microcircuitry itself.

**Plasticity in the brain** The precise mechanisms for plasticity in the brain are still under debate. Popular models include *Hebbian learning* and its variant *spike time dependent plasticity*, both of which adjust synapse strengths based on local firing history. Although these rules are usually modeled via additive updates, it has been suggested that multiplicative updates may better explain the available data—both in terms of the induced stationary distribution of synapse strengths, and also time-dependent observations [40–42]. For example, Loewenstein et al. [43] image dendritic spines in the mouse auditory cortex over several weeks. Upon finding that changes in spine size are "proportional to the size of the spine", the authors suggest that multiplicative updates are at play.

In contrast to these studies that concentrate on matching candidate update rules to available data—and thus probing *how* the brain learns—this paper has focused on using perturbation theory and calculus to link update rules to the stability of network function. This is a complementary approach that may shed light on *why* the brain learns the way it does—paving the way, perhaps, for computer microarchitectures that mimic it.

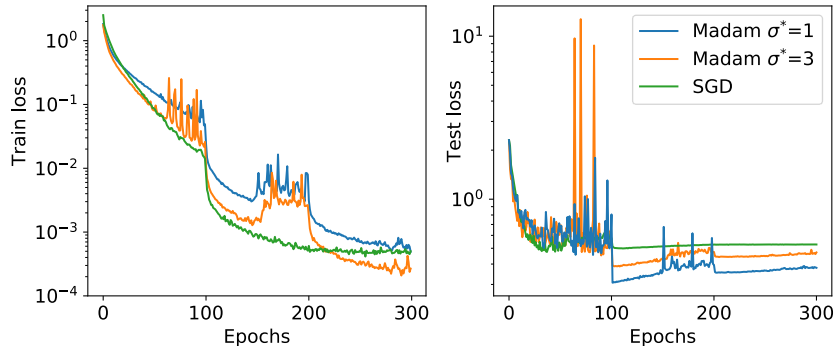

Figure 3: Convergence speed and effect of weight clipping. We plot the convergence curves of SGD and Madam on CIFAR-10. Two different weight clipping thresholds $\sigma^*$ are shown for Madam. Reducing the clipping threshold $\sigma^*$ can be seen to both stabilise and regularise the test performance.

## 8 Limitations and future work

There are two aspects of this work that we would like to understand better. The first is the role of the weight clipping threshold $\sigma^*$ (see Algorithm 1). Figure 3 shows a comparison between an SGD baseline and Madam with different weight clipping thresholds. As can be seen, a reduced clipping threshold seems to both *stabilise* and *regularise* the test loss. The stabilising role of weight clipping may be understood as follows: at the end of training, a learning algorithm can overfit the training set by simply scaling up the learnt weights (amplifying confidence on its current predictions). Since Madam can increase the weights *exponentially* (via compound growth over a number of iterations), this effect is particularly unstable and motivates explicit weight clipping. Still, needing to tune this clipping threshold is undesirable, and it would be better if there were a way to avoid it.

The second is the slight deterioration of Madam compared to Adam and SGD in Table 1. It is not immediately clear where this deterioration comes from. One possibility is that, since Madam freezes the sign pattern of weights from initialisation, the network is not expressive enough to represent the very best solutions. A workaround to this potential problem could involve finding a better way to initialise the sign pattern of the weights at the start of training. More generally, it is possible that we are missing some key idea that would close this performance gap.

### Broader impact

This paper proposes that multiplicative update rules may be better suited to the compositional structure of neural networks than additive update rules. It concludes by discussing possible implications of this idea for chip design and neuroscience. The authors believe the work to be fairly neutral in terms of propensity to cause positive or negative societal outcomes.

### Acknowledgements and disclosure of funding

The authors would like to thank the anonymous reviewers for their helpful comments.

JB was supported by an NVIDIA fellowship. The work was partly supported by funding from NASA.

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
