[Supplementary Material]

# Appendix A  Proofs

**Lemma 1.** *Consider a continuously differentiable function $\mathcal{L} : \mathbb{R}^n \to \mathbb{R}$ that maps $W \mapsto \mathcal{L}(W)$. Suppose that parameter vector $W$ decomposes into $L$ parameter groups: $W = (W_1, W_2, ..., W_L)$, and consider making a perturbation $\Delta W = (\Delta W_1, \Delta W_2, ..., \Delta W_L)$. Let $\theta_k$ measure the angle between $\Delta W_k$ and negative gradient $-g_k(W) := -\nabla_{W_k} \mathcal{L}(W)$. Then:*

$$\mathcal{L}(W + \Delta W) - \mathcal{L}(W) \leq -\sum_{k=1}^{L} \|g_k(W)\|_F \|\Delta W_k\|_F \left[ \cos \theta_k - \max_{t \in [0,1]} \frac{\|g_k(W + t\Delta W) - g_k(W)\|_F}{\|g_k(W)\|_F} \right].$$

*Proof.* By the fundamental theorem of calculus,

$$\mathcal{L}(W + \Delta W) - \mathcal{L}(W) = \sum_{k=1}^{L} \left[ g_k(W)^T \Delta W_k + \int_0^1 \left[ g_k(W + t\Delta W) - g_k(W) \right]^T \Delta W_k \, \mathrm{d}t \right].$$

The result follows by replacing the first term on the righthand side by the cosine formula for the dot product, and bounding the second term via the integral estimation lemma. $\square$

**Modelling assumption 1.** [Deep relative trust] Consider a neural network with $L$ layers and parameters $W = (W_1, W_2, ..., W_L)$. Consider parameter perturbation $\Delta W = (\Delta W_1, \Delta W_2, ..., \Delta W_L)$. Let $g_k(W) := \nabla_{W_k} \mathcal{L}(W)$ denote the gradient of the loss. Then the gradient breakdown is bounded by:

$$\frac{\|g_k(W + \Delta W) - g_k(W)\|_F}{\|g_k(W)\|_F} \leq \prod_{l=1}^{L} \left( 1 + \frac{\|\Delta W_l\|_F}{\|W_l\|_F} \right) - 1 \qquad \text{(for } k = 1, ..., L\text{)}.$$

**Theorem 1.** *Let $\mathcal{L}$ be the continuously differentiable loss function of a neural network of depth $L$ that obeys deep relative trust. For $k = 1, ..., L$, let $0 \leq \gamma_k \leq \frac{\pi}{2}$ denote the angle between $|g_k(W)|$ and $|W_k|$ (where $|\cdot|$ denotes the elementwise absolute value). Then the multiplicative update in Equation 3 will decrease the loss function provided that:*

$$\eta < (1 + \cos \gamma_k)^{\frac{1}{L}} - 1 \qquad \text{(for all } k = 1, ..., L\text{)}.$$

*Proof.* Using the gradient reliability estimate from deep relative trust, we obtain that:

$$\max_{t \in [0,1]} \frac{\|g_k(W + t\Delta W) - g_k(W)\|_F}{\|g_k(W)\|_F} \leq \max_{t \in [0,1]} \prod_{l=1}^{L} \left( 1 + \frac{\|t\Delta W_l\|_F}{\|W_l\|_F} \right) - 1 \leq \prod_{l=1}^{L} \left( 1 + \frac{\|\Delta W_l\|_F}{\|W_l\|_F} \right) - 1.$$

Descent is guaranteed if the bracketed terms in Lemma 1 are positive. By the previous inequality, this will occur provided that:

$$\prod_{l=1}^{L} \left( 1 + \frac{\|\Delta W_l\|_F}{\|W_l\|_F} \right) < 1 + \cos \theta_k \qquad \text{(for all } k = 1, ..., L\text{)} \tag{6}$$

where $\theta_k$ measures the angle between $\Delta W_k$ and $-g_k(W)$. For the update in Equation 3,

$$W + \Delta W = W \odot (1 - \eta \operatorname{sign} W \odot \operatorname{sign} g(W)).$$

Therefore the perturbation is given by $\Delta W = -\eta |W| \odot \operatorname{sign} g(W)$. For this perturbation, $\|\Delta W_*\|_F / \|W_*\|_F = \eta$ for any possible subset of weights $W_*$. Also, letting $\measuredangle(\cdot, \cdot)$ return the angle between its arguments, $\theta_k$ and $\gamma_k$ are related by:

$$\begin{aligned}
\theta_k &:= \measuredangle(\Delta W_k, -g_k(W)) \\
&= \measuredangle(-\eta |W_k| \odot \operatorname{sign} g_k(W), -g_k(W)) \\
&= \measuredangle(|W_k| \odot \operatorname{sign} g_k(W), g_k(W)) \\
&= \measuredangle(|W_k|, |g_k(W)|) \\
&=: \gamma_k.
\end{aligned}$$

Substituting these two results back into Equation 6 and rearranging, we are done. $\square$

Table 3: Complete results for FP32 Madam. We quote top-1 error, FID [36] and perplexity for the classifiers, GAN and transformer respectively. Lower is better in all cases. The mean result is quoted with a range based on three repeats.

*Note: all Madam runs use initial learning rate $\eta = 0.01$.*
*For all algorithms, $\eta$ is decayed by 10 when the loss plateaus.*

| Dataset | Adam $\eta$ | Adam train | Adam test | SGD $\eta$ | SGD train | SGD test | Madam $\eta$ | Madam train | Madam test |
|---|---|---|---|---|---|---|---|---|---|
| CIFAR-10 | 0.001 | $0.01 \pm 0.01$ | $6.6 \pm 0.2$ | 0.1 | $0.01 \pm 0.01$ | $8.2 \pm 1.3$ | 0.01 | $0.01 \pm 0.01$ | $7.8 \pm 0.2$ |
| CIFAR-100 | 0.001 | $0.02 \pm 0.01$ | $29.8 \pm 0.4$ | 0.1 | $0.03 \pm 0.01$ | $29.1 \pm 0.2$ | 0.01 | $2.35 \pm 0.08$ | $30.2 \pm 0.1$ |
| ImageNet | 0.01 | $19.8 \pm 0.2$ | $26.7 \pm 0.3$ | 0.1 | $17.7 \pm 0.1$ | $24.1 \pm 0.1$ | 0.01 | $25.4 \pm 0.1$ | $28.9 \pm 0.1$ |
| cGAN | 0.0001 | $23.1 \pm 0.8$ | $23.9 \pm 0.9$ | 0.01 | $34 \pm 1$ | $34 \pm 1$ | 0.01 | $19 \pm 2$ | $19 \pm 2$ |
| Wikitext-2 | 0.0001 | $109.8 \pm 0.5$ | $173.4 \pm 0.9$ | 1.0 | $149.9 \pm 0.2$ | $169.6 \pm 0.6$ | 0.01 | $126.9 \pm 0.2$ | $173.3 \pm 0.6$ |

## Appendix B    Experimental details

**FP32 Madam**    The initial learning rate was set to $\eta = 0.01$ across all tasks, while the weight clipping threshold $\sigma^*$ was tuned over the range 1 to 5. The learning rate schedule was set specific to each task, although the general strategy was to decay the learning rate upon plateau of the loss.

**$B$-bit Madam**    For 12-bit Madam, the base precision $\eta_0$ was set to 0.001 across all tasks. To reduce precision to 10 bits and 8 bits, we maintained the same dynamic range as for 12 bits by increasing the base precision $\eta_0$. For example, in all the image classification experiments, the initial learning rate $\eta$ was set to 0.016 for all bit widths, while the learning rate was decayed to 0.001 in the 12-bit experiments, 0.004 in the 10-bit experiments and never decayed in the 8-bit experiments. In the GAN and transformer experiments, the initial learning rate $\eta$ was set to 0.01. The difference between an initial learning rate of 0.016 and 0.01 was not highly significant—the learning rate of 0.016 was chosen to be compatible with code that decayed the learning rate by powers of two. The weight clipping threshold $\sigma^*$ was set to the value used in FP32 Madam for each task.

**CIFAR-10 classification**    We evaluated the different optimisers on the CIFAR-10 dataset [44] using a Resnet-18 model [45]. CIFAR-10 consists of 60,000 images in 10 different classes. For simplicity, we switched off the learnable affine parameters in batch norm. The network was trained for 300 epochs, and we used a fixed learning rate decay schedule that decayed every 100 epochs.

**CIFAR-100 classification**    The CIFAR-100 dataset is similar to CIFAR-10, but contains 100 classes containing 600 images each [44]. Again, we used a Resnet-18 model with the batch norm affine parameters switched off. The other hyperparameters were chosen in the same way as for CIFAR-10, including the epoch budget and learning rate decay strategy.

**ImageNet classification**    The ILSVRC2012 ImageNet dataset consists of 1.2 million images belonging to 1,000 classes [46]. We used a Resnet-50 network architecture to benchmark the optimisers [45]. Again, we switched off the affine parameters in batch norm for simplicity. The network was trained for 90 epochs. We applied a learning rate warm-up for the first 5 epochs of training, and decayed the learning rate after every 30 epochs.

**CIFAR-10 GAN**    We trained a class-conditional generative adversarial network (cGAN) using a custom implementation of the BigGAN architecture [47], to learn to generate the CIFAR-10 dataset [44]. The same learning rate was used in both the generator and discriminator. The networks were trained for 120 epochs. The learning rate was decayed after 100 epochs of training, by a factor of 10 in the FP32 experiments and to the base precision in the $B$-bit experiments. Bias parameters were initialised from a Normal distribution with mean zero and variance 0.01.

**Wikitext-2 transformer**    WikiText-2 is a language modeling dataset that contains over 100 million tokens extracted from Wikipedia [48]. We trained a transformer network architecture [49] that was smaller than the transformers that are typically used for this task, explaining the general degradation of results compared to the state of the art for this dataset. The network was trained for 20 epochs with a single learning rate decay at epoch 10. In 12-bit Madam, decaying the learning rate to 0.005 rather than to the base precision of 0.001 slightly improved the results.