[Reviews · NeurIPS 2020]

Review 1

Summary and Contributions: This paper analyzes the multiplicative update (or exponential gradient) methods for compositional objective functions. Under the so-termed deep relative trust assumption, the descent lemma has been established analogous to the gradient descent under the smooth assumption.

Strengths: 1. The idea of applying the multiplicative update to solve the compositional objective functions is new and may have some impact. 2. The development of Adam version of the multiplicative update is potentially useful.

Weaknesses: 1. The connection between the proposed multiplicative update with neural computation seems redundant or even distracting. 2. The theoretical analysis is on the weak side. It is not clear where the solution will converge to even if the assumption is satisfied and the descent lemma is established. Is it a stationary point, local min or global min? 3. The performance gain of the new algorithm relative to SGD and Adam is at most marginal. The theory and algorithm may be useful, but it needs a killer application.

Correctness: The results seem to be correct.

Clarity: The paper is well-written, and it is pleasant to read.

Relation to Prior Work: The relation to prior work is sufficient.

Reproducibility: Yes

Additional Feedback: 1. The algorithm and the analysis relies on the new assumption of deep relative trust, it is not clear when it is satisfied. Is it for any compositional function? For NN, it is satisfied for any structure and any activation functions? 2. What is the advantage and merit of the new analysis relative to the existing analysis of multiplicative update methods such as exponential gradient. ========= After Rebuttal ================== I have read the rebuttal from the authors. My score remains the same. Thanks for clarifying some points.


Review 2

Summary and Contributions: The paper proposes a novel weight update algorithm which is multiplicative in weight space (thus additive in log-weight space). Some theoretical justification is presented, followed by experimental results, which show that the method may work well over a range of problems without tuning the learning rate (this is in contrast to SGD and ADAM, which are the other methods benchmarked). A simple, low precision variant of the method is shown to perform reasonably well.

Strengths: In my opinion there are two interesting (closely related) ideas in this work. The first is the multiplicative update rule, and the second is the numerical representation with only exponent bits and no mantissa. Although multiplicative update rules are a well established technique in 'classical' machine learning, it is interesting to see some investigation of how well these techniques work in a more modern setting. I found the theoretical section of the paper (section 3) convincing and reasonably straightforward to understand.

Weaknesses: I was not totally convinced by the experiments section, and have questions about that section and some more general questions which the authors might address: 1. The way that Figure 1 is laid out suggests that it is appropriate to compare the three algorithms over the same set of values of eta. Can the authors justify this? It seems to me that the meaning of eta in the Madam algorithm is different to its meaning in SGD and Adam (it's effectively a coincidence that these different hyper-parameters share a name). What happens if you evaluate Madam over a denser grid of eta values and then zoom in the x axis of the left hand plot? 2. The value of the transformer, on the wikitext-2 task, for SGD and Madam, seems very high. I'm not directly familiar with this task, and am comparing to values from https://paperswithcode.com/sota/language-modelling-on-wikitext-2 where it looks like all papers achieved a loss <= 100. Perhaps the authors are using a different unit of measurement? 3. You describe your algorithm as 'competetive' on all baselines, but it appears to be significantly worse (4.8%) than SGD on the ImageNet task. Could you change your description to clarify that it is competetive on *almost* all tasks. 4. You mention that you had to alter the initialization of bias terms because PyTorch defaults to initializing them to zero, which makes the multiplicative update rule ineffective. Does the theory presented in Section 3 apply to bias terms? It seems like it is based on an approximation in which neural net layers involve multiplication with weights, rather than addition. Could you include in the paper some discussion of whether the multiplicative rule is appropriate for bias values, and if it is not can you suggest an alternative way to handle biases (maybe just fall back to an existing algorithm?). 5. Regarding the sentence beginning 'Surprisingly' on line 225: is this a coincidence? Can you be more explicit about what you are suggesting here or else remove the sentence? 6. On line 130, the value you give for gamma (=pi / 4) seems arbitrary. It seems like it wouldn't be too hard to at least simulate and find the average value over random vectors W and g(W).

Correctness: I think that the theoretical claims are correct, although I haven't had enough time to check them really carefully. The empirical methodology seems correct, with the caveats/questions mentioned above.

Clarity: The paper is very clearly written, I didn't find any grammatical errors or typos and everything was easily readable and made sense. One minor point: the vertical bar notation notation used on line 127, for the absolute values of the vectors g(W) and W, is not defined anywhere.

Relation to Prior Work: Yes, the authors give a good summary of prior work and its relation to theirs. They might want to add a citation to the Weight Normalization paper: https://papers.nips.cc/paper/6114-weight-normalization-a-simple-reparameterization-to-accelerate-training-of-deep-neural-networks, which I believe is related but not mentioned.

Reproducibility: Yes

Additional Feedback:


Review 3

Summary and Contributions: Based on the recently introduced concept of "deep relative trust", the authors 1) propose and analyze a multiplicative learning rule, 2) benchmark a practical instantiation of this rule (dubbed Madam), along with low-precision implementations of this algorithm and 3) discuss connections between multiplicative learning rules and neuroanatomical observations.

Strengths: Significance: 1) One of the main practical contributions of this paper, which is a multiplicative optimizer that's not sensitive to careful tuning of learning rate, can be of great practical significance. 2) The compelling results obtained with the low-bit MADAM might have important implications on hardware design and large-scale optimization. 3) I find the ideas presented in the paper (including the idea of using multiplicative update rules for compositional function learning) interesting, and expect that it will spawn future research in this direction. Soundness of the claims: 1) Mathematical analysis of the proposed method: I've checked the mathematical analyses presented in the paper (including the proofs in the appendix) and I think all the results are correct and well presented/justified. 2) Empirical evaluation: The experiments serve 3 purposes: 1) demonstrate that the same learning rate is optimal for 5 different tasks (cifar and imagenet classification, conditional gan training and language modelling) for MADAM 2) the performance of the optimizer is competitive with some go-to optimizers like Adam and SGD and 3) Quantify how much performance degrades (if at all) in the low bit setting. The results look compelling on all fronts. Especially the fact that the performance of MADAM is better than the competitors for the conditional GAN training is encouraging, as the learning rate and schedule is often a critical in stabilizing adversarial training problems. Novelty: To the best of my knowledge, both the analysis and the proposed optimizer are novel. Relevance: The topic and contributions of this paper is definitely relevant to the NeurIPS community.

Weaknesses: Empirical evaluation: 1) I'd strongly recommend the authors to dedicate a substantial section in the appendix detailing the experimental setup for all the experiments. Even though the codebase used for some of these experiments are released, it is very difficult for the reviewers to reverse engineer the details of the experiments from this. 2) I didn't see any analysis of how fast Madam converges. This being an extremely important consideration for practitioners, I consider the lack of such results a weakness. It would be great if the authors add a section discussing this. It would also be great to see a (albeit brief) discussion]of the computational cost incurred by the matrix exponentiation operation in the algorithm. Proposed optimizer (MADAM): 1) The last step of the MADAM algorithm (Algorithm 1 in the paper) introduces weight clipping. Section 5 of the paper mentions that lightly tuning this value leads to improvements. I'm concerned that this operation will constrain the expressive power of the networks being trained and lead to weaker results than they could have been without this operation. For a given network architecture, weight clipping imposes a Lipschitz bound on the function the network can represent. I'm also concerned that this operation might be suppressing some other kind of instability that the algorithm displays. In this regard, I have a few questions to the authors: 1) Why exactly is this operation needed? Does the algorithm behave unstably without it? If so, what's the nature of this unstability? 2) You mention that this operation is comparable to the applying weight decay in SGD. Did you try actually applying weight decay instead of this clipping operation, or some more benign type of regularization? 3) If you analyze a converged network, what percentage of the weights are at the maximum/minimum values allowed by clipping? 4) Is the best \sigma_{max} roughly the similar across different architectures/tasks? 5) As expressed by the authors, a search of \sigma_{max} constitutes some kind of regularization hyperparameter tuning. In section 5 of the paper (i.e. Madam benchmarking experiments), did you also tune some kind of regularization hyperparameter for the baselines, such as by weight decay, or even by the same weight clipping operation?

Correctness: Soundness of the claims: 1) Mathematical analysis of the proposed method: I've checked the mathematical analyses presented in the paper (including the proofs in the appendix) and I think all the results are correct and well presented/justified. 2) Empirical evaluation: I'd still strongly recommend the authors to dedicate a substantial section in the appendix detailing the experimental setup for all the experiments. Even though the codebase used for the experiments are released (which is extremely helpful), it is difficult to faithfully reverse engineer the details of the experiments from this. 3) Connections to neuroanatomy: It could be considered a bit speculative to attribute the similarity between the dynamic range of the synapse strengths of mice brain and the dynamic range of weights in B=12 bit MADAM to anything other than chance. However, the authors do mention this only as a surprising connection and don't put too much weight on this.

Clarity: Yes - the paper is easy to follow and understand, and the results are clearly presented.

Relation to Prior Work: As far as my knowledge goes, the paper has a strong related works section that situates it properly in the existing literature.

Reproducibility: Yes

Additional Feedback: 1) Page 3, line 100: Could you explain the phrase "for every subset W_{*} of weights W" a bit more in detail, hopefully using equations? 2) How would you apply MADAM to architectures like DenseNet where the notion of a layer is not as well defined? Would the "dense blocks" be considered as "layers"? Nitpick: 1) It might help mentioning the \eta is a positive scalar somewhere earlier, in page 3. 2) It might be interesting to the readers to emphasize that the result of Theorem 1 holds for all weight initialization procedures. === After author feedback === Thank you for your clarifications in your response. Since weight clipping seems to be an important component of the MADAM algorithm and has a large influence on the training dynamics by both stabilizing learning and regularizing the network, I believe a more thorough discussion of this could improve the paper. Also considering some of the other points raised by the other reviewers, I’m inclined to slightly lower my score. However, I still think this is a good paper making some interesting connections between compositional function learning, multiplicative learning algorithms and optimizers that are suitable for low-precision hardware that many in NeurIPS community might benefit from.


Review 4

Summary and Contributions: The submitted paper proposes an optimization algorithm called Madam using insights from multiplicative weight updates. Under some modelling assumptions (called "deep relative trust assumption"), the paper shows that Madam removes the need for careful learning rate tuning. Moreover, since the algorithm can represent the network’s weight in the logarithmic number system, it can be easily adapted to train a compressed neural network. At last, the paper makes a connection between Madam and synapses in the brain. Empirical evaluations were done to show that the proposed method has a competing performance compared to the baseline results (SGD and Adam).

Strengths: - Overall, the paper is well-organized and the derivation looks correct. - The main idea is well-motivated and interesting. To my knowledge, an optimization algorithm built upon multiplicative update with insights from synapses was not proposed before and thus novel. It can potentially be impactful. - While limited, the empirical experiments hint that Madam can have competing performance compared to SGD and Adam. - The work is relevant to the Neurips community.

Weaknesses: (After rebuttal) Thank you for the author's feedback. While some of my concerns were addressed, I still believe that there needs to be more (empirical) works to verify deep relative trust is a sufficiently good model. As the authors clarified the theory on deep relative trust and some experimental details, I increased my score to 6. -------- - The paper lacks a discussion on deep relative trust assumptions. I am not sure under what conditions (e.g. loss, activations, architecture) the modelling assumptions are realistic on neural networks. - The writings (especially method sections) can be improved by clearly highlighting the assumptions and the motivations. - The experiments are not comprehensive. The plots for showing the convergence for Madam and the baselines would be useful. I feel like some experiments showing the trajectory of Madam would be helpful to capture the insight as well. Moreover, a comparison with other compression methods would help to understand the strength of Madam. - It still requires tuning some hyperparameters. The empirical results do not show how sensitive these hyperparaeters are. I would be interested to see how Madam performs on different meta-parameters (e.g. \sigma, \sigma^*), along with the discussion on their impact. - A more detailed explanation of the experiment set-up would be useful. E.g. is it using a fixed learning rate throughout training?

Correctness: The claims, proofs, and methods look correct. While limited, the empirical methodology presented in the paper looks mostly correct. It would be helpful to show the performance for each SGD and Adam in Table 1. Also, in Figure 1, is the error for testing or training dataset?

Clarity: While the paper is well-organized, the writings can be improved in method sections. It would be helpful if the authors can include a more detailed explanation and discussion (also highlighting major assumptions and motivations).

Relation to Prior Work: The paper is well-referenced. Also, it clearly discusses how it differs from previous learning algorithms. Hence, the relation to prior work is sufficient.

Reproducibility: Yes

Additional Feedback: Minor comments / Questions: * In equation (1), W_k should be defined. * In page 3, \eta is not defined. * Missing commas in several places (e.g. line 112, 154, 241, Table 1). * In line 116, where does 10^18 come from? * In the equation above lines 157 and 205, the indices should be a set. * What does it mean by O(0.02) or O(+- 1)? * Shouldn't the first Lemma be referenced [1]? [1] Bernstein, J., Vahdat, A., Yue, Y., & Liu, M. Y. (2020). On the distance between two neural networks and the stability of learning. arXiv preprint arXiv:2002.03432.

[Author Response · NeurIPS 2020]

**All reviewers & AC**   We thank the reviewers for their thoughtful feedback. The reviewers unanimously agree that the central idea of the work is novel, and reviewer 5 states that the work might have "important implications on hardware design and large-scale optimization". We want to assure the reviewers that we will heed their advice—for example:

   1) We will add a substantial section to the appendix detailing our complete experimental setup.

   2) We will add more discussion of the major assumptions and motivations underlying deep relative trust.

Multiplicative updates depart from the best practices of deep learning that were refined over many research iterations. Though our early results are promising, we believe that more research effort is needed to improve our new techniques. Sharing this work with the NeurIPS community would help to start this conversation.

<div align="center">***</div>

**When is deep relative trust satisfied?  (R1, R6)**   For a rigorous derivation of the deep relative trust distance function, please see [1, Theorem 1]. The proof holds for multilayer perceptrons with full rank weight matrices and leaky relu nonlinearity. While deep relative trust is unlikely to hold *exactly* for arbitrary neural networks or compositional functions, we contend that it is a better model of neural networks than its alternatives (e.g. Lipschitz gradients).

[1] Bernstein et al. 2020. *On the distance between two neural networks and the stability of learning.* arXiv:2002.03432.

**Comparison to exponential gradient (R1)**   The exponential gradient algorithm is classically derived via mirror descent with a KL-divergence distance function. This leads to *unbounded* relative updates of the form $\exp(\eta g)$. Our analysis, on the other hand, leads to *bounded* relative updates of the form $\exp(\eta \operatorname{sign} g)$. Bounded and unbounded relative updates have substantially different properties. More broadly, our analysis is tailored to neural networks via the deep relative trust assumption—in contrast, KL-divergence seems to have no obvious connection to neural networks.

**Strength of the theory (R1)**   Our empirical results question the suitability of the modelling assumptions used to prove many existing theory results in deep learning. We therefore believe our work to have strong theoretical significance despite only proving one simple result in this paper. That said, we should mention that Theorem 1 *does* entail convergence to stationary points, provided no weight is initialised to zero. To see this, note that the weights stay non-zero under a multiplicative update, so $g(W) \neq 0 \implies \cos\gamma > 0 \implies$ descent is possible by Theorem 1.

**Convergence plots (R5, R6)**   Though we did not include this in the paper, Madam converges at a very similar speed to Adam and SGD. See the inset figure for a comparison on CIFAR-10 classification. Note that reduced $\sigma^*$ both stabilises *and* regularises the test loss.

**Discussion on weight clipping via $\sigma^*$ (R5)**   At the end of training, a learning algorithm can overfit the training set by simply scaling up the learnt weights (amplifying confidence on its current predictions). Since Madam can increase the weights *exponentially*, this effect was particularly unstable and led us to introduce weight clipping. Afterward we discovered that tuning $\sigma^*$ had a regularizing effect. The best $\sigma^*$ was fairly consistently between 1 and 3 times the Xavier init scale. We did not experiment thoroughly with alternative regularizers in Madam. The baselines we compare against are heavily tuned—for example, SGD on Imagenet uses a weight decay value of $10^{-4}$.

**Reviewer 1 miscellanea**   On a *killer app*: the tuning gains and easy low bit width training may be highly significant.

**Reviewer 5 miscellanea**   We would like to clarify one potential misunderstanding. Madam uses elementwise exponentiation (not matrix exponentiation)—thus the iteration cost of Madam is very similar to Adam. When ported to log number system hardware, Madam should be significantly faster than Adam. Also, since Madam is purely elementwise, applying Madam to DenseNet is simple and does not require defining a notion of a layer. Now let us clarify the phrase "for every subset $W_*$ of weights W"—let's say a network has $N$ weights $W = [W_1, W_2, ..., W_N]$. Now take any subset of the indices, e.g. $S = \{2, 4, 5\}$, and construct a new weight vector $W_*$ based on $S$—in this case $W_* := [W_2, W_4, W_5]$. Then after a multiplicative update of size $\eta$, the relative change in the vector 2-norm of *any* such $W_*$ is $\eta$.

**Reviewer 6 miscellanea**   Figure 1 shows test set results. A Hessian with $10^{18}$ entries comes from assuming a neural network with $10^9$ weights (e.g. GPT-2). By O(0.01) we mean the same order of magnitude as 0.01—specifically, one coauthor used $\eta = 0.016$ in some low-precision experiments so that they could decay to the base precision by dividing by 2. Finally, here are the complete FP32 results. The metrics are classification error $(100 - \text{accuracy})$ for the classifers, FID score for the cGAN and perplexity for the transformer.

| Dataset | Adam $\eta$ | Adam train | Adam test | SGD $\eta$ | SGD train | SGD test | Madam $\eta$ | Madam train | Madam test |
|---|---|---|---|---|---|---|---|---|---|
| CIFAR-10 | 0.001 | $0.01 \pm 0.01$ | $6.6 \pm 0.2$ | 0.1 | $0.01 \pm 0.01$ | $8.2 \pm 1.3$ | 0.01 | $0.01 \pm 0.01$ | $7.8 \pm 0.2$ |
| CIFAR-100 | 0.001 | $0.02 \pm 0.01$ | $29.8 \pm 0.4$ | 0.1 | $0.03 \pm 0.01$ | $29.1 \pm 0.2$ | 0.01 | $2.35 \pm 0.08$ | $30.2 \pm 0.1$ |
| ImageNet | 0.01 | $19.8 \pm 0.2$ | $26.7 \pm 0.3$ | 0.1 | $17.7 \pm 0.1$ | $24.1 \pm 0.1$ | 0.01 | $25.4 \pm 0.1$ | $28.9 \pm 0.1$ |
| cGAN | 0.0001 | $23.1 \pm 0.8$ | $23.9 \pm 0.9$ | 0.01 | $34 \pm 1$ | $34 \pm 1$ | 0.01 | $19 \pm 2$ | $19 \pm 2$ |
| Wikitext-2 | 0.0001 | $109.8 \pm 0.5$ | $173.4 \pm 0.9$ | 1.0 | $149.9 \pm 0.2$ | $169.6 \pm 0.6$ | 0.01 | $126.9 \pm 0.2$ | $173.3 \pm 0.6$ |

[Meta-Review · NeurIPS 2020]

This is a good paper which combines insights from optimization, hardware, and neuroscience to give a multiplicative weight update for neural nets. It seems worthwhile to try out multiplicative updates in the context of modern architectures, and this paper seems to have made them competitive with existing optimizers, in a way that allows lower-precision computation (as low as 8 bits). As far as I can tell, there isn't a clear advantage for current hardware, but this serves as a good proof-of-concept that could help inform future hardware design. While no particular insight is particularly deep, everything is combined in an interesting and cohesive way, so the reviewers and I think this paper is definitely above the bar for acceptance. I encourage the authors to account for the reviewers' feedback in the camera ready version.